# Particle Size-Dependent Component Separation Using Serially Arrayed Micro-Chambers

**DOI:** 10.3390/mi14050919

**Published:** 2023-04-24

**Authors:** Mitsuhiro Horade, Ryuusei Okumura, Tasuku Yamawaki, Masahito Yashima, Shuichi Murakami, Tsunemasa Saiki

**Affiliations:** 1Department of Mechanical Systems Engineering, National Defense Academy of Japan, 1-10-20 Hashirimizu, Yokosuka 239-8686, Japan; 2Osaka Research Institute of Industrial Science and Technology, 2-7-1 Ayumino, Izumi 594-1157, Japan; 3Hyogo Prefectural Institute of Technology, 3-1-12 Yukihira, Suma, Kobe 654-0037, Japan

**Keywords:** component separation, micro-manipulation, PDMS, chamber array, behavior analysis

## Abstract

The purpose of this research was to enable component separation based on simple control of the flow rate. We investigated a method that eliminated the need for a centrifuge and enabled easy component separation on the spot without using a battery. Specifically, we adopted an approach that uses microfluidic devices, which are inexpensive and highly portable, and devised the channel within the fluidic device. The proposed design was a simple series of connection chambers of the same shape, connected via interconnecting channels. In this study, polystyrene particles with different sizes were used, and their behavior was evaluated by experimentally observing the flow in the chamber using a high-speed camera. It was found that the objects with larger particle diameters required more time to pass, whereas the objects with smaller particle diameters flowed in a short time; this implied that the particles with a smaller size could be extracted more rapidly from the outlet. By plotting the trajectories of the particles for each unit of time, the passing speed of the objects with large particle diameters was confirmed to be particularly low. It was also possible to trap the particles within the chamber if the flow rate was below a specific threshold. By applying this property to blood, for instance, we expected plasma components and red blood cells to be extracted first.

## 1. Introduction

This study reports a component separation method which uses an easily processable microfluidic device comprising an array of submillimeter-sized chambers and interconnecting channels. The proposed method is portable, inexpensive, and battery-free. Centrifugation is a typical separation method that uses the centrifugal force to separate substances having different specific gravities [1,2,3,4]. For instance, in the case of blood, the erythrocytes settle at the bottom of the tubular container, and the plasma, which is a liquid component, is collected as a supernatant. The method proposed in this study is similar to the centrifugal method; however, the separation performance might be inferior to that resulting from the use of a macro centrifuge, which is the backbone of component separation. In this study, we analyzed the behavior of polystyrene particles under various conditions using a syringe pump with a controllable flow rate and high-speed camera for observing and evaluating the separation process. Based on the experimental results, the separation performance and potential for future application development is discussed.

Numerous studies have been conducted on component separation using microfluidic devices and microelectromechanical systems (MEMSs). Depending on the size of the object, separation can be achieved through physical trapping or by moving the object using an external force. As an example of physical trapping, a mechanism to trap objects inside a microfluidic device has been reported; it comprises silicon nanowires, pillars, and microchambers arranged inside a channel [5,6,7,8,9,10,11,12,13,14]. However, there are cases where the processing time increases in some devices and cases where the materials selectivity is limited. As an example of methods using an external force [15,16,17,18], actuators have been integrated so that centrifugal force is applied inside the microfluidic device [19,20,21,22]. In addition, electrode patterning is necessary to generate a dielectric force and perform signal control using a function generator [23,24,25,26]. However, using an actuator requires a power source and time to develop the device that forms the electrodes. In this study, we proposed a separation method based only on the flow rate control of a syringe pump when introducing the suspension to be separated into the microfluidic device.

This method can be implemented by controlling only the flow rate; therefore, it could be used in cases where quick separation is required in an on-site environment. In addition, the channel can be formed from polydimethylpolysiloxane (PDMS) and glass using versatile soft lithography; therefore, it can be developed with minimum equipment. Moreover, the separation performance might be superior to that in other methods. For instance, with a method using the dielectric force, even if two or more objects with the same specific gravity are included, only a specific object can be separated by optimizing the frequency [27,28]. In the proposed method, only substances with different sizes can be separated; however, it may be used in applications such as easy concentration adjustment and collection of liquid-only components. Therefore, the present research intended to develop a fluidic separation device that could conduct separation tests according to the flow rate and size of the object, and considered potential applications based on the final results. Herein, Section 2 describes the device design and experimental conditions. Section 3 describes the experimental results. Section 4 presents the discussion. Finally, Section 5 reports the summary of this research.

## 2. Materials and Methods

### 2.1. Design Guidelines

The microfluidic device used in this study was comprised of an array of chambers connected in series through channels. In previous studies, it was confirmed that the larger the object, the lower the flow velocity in a chamber with a large spatial volume [29]. It was expected that this research would utilize this knowledge and lead to applications in component separation according to particle size, concentration adjustment, and cell trapping in the flow channel. Considering the ease of observation under a microscope, the design of the chamber and channel was simplified from that in the previous research.

Figure 1 shows a schematic of the microfluidic device fabricated in this study. An inlet well, an outlet well, and 50 chambers were placed and connected in series. Considering the adhesion to the versatile 25 mm × 75 mm slide glass, the PDMS mold was made to be within 25 mm × 25 mm. Therefore, the chambers were arranged in a 5 × 10 grid pattern, and the chambers were interconnected through channels with line widths shorter than those of the chambers. For example, when a blood sample was introduced as shown in Figure 1, the plasma (which is a liquid component), small-sized erythrocytes, and large-sized white blood cells flowed in this order of decreasing speed. The right side of Figure 1 is an enlarged image of the chamber. The large-sized white blood cells slowed down the flow in the chamber. The effect per chamber was small; however, if multiple chambers were arranged in series, the effect increased accordingly. In this study, we quantitatively evaluated the effect of each particle in a chamber and proposed a design guideline to easily observe the behavior of particles in the chamber. The specific dimensions are shown in Figure 2. The chamber size was 800 μm × 800 μm. Adjacent chambers were connected by 200 μm × 2200 μm channels and were separated by 600 μm. However, due to the grid-like arrangement, some parts were connected by channels with lengths of 2200 μm or more. Both the chamber and channel were 100 µm in height. Although the size varied depending on the cell type, we set the above dimensions to observe not only cells of 10–15 μm but also the behavior of larger objects of 40–50 μm in the channel.

### 2.2. Fabrication Process

Figure 3 depicts the fabrication process. Using a diamond cutter, a silicon wafer with a thickness of 525 µm was cut into 2.5 cm squares, which were ultrasonically cleaned for 10 min to remove as many silicon fragments as possible. Ultrasonic cleaning was performed twice with pure water and once with ethanol, and the liquid was changed three times in total. Each silicon substrate was spin-coated with SU-8 3005 photoresist at 1000 rpm. Subsequently, it was soft-baked at 90 °C for 45 min. A contact exposure was performed at a dose of 100 mJ/cm^2^ using a mask aligner (MA-6; SUSS MicroTec, Munich, Germany). Afterwards, a post-exposure bake was performed at 95 °C for 7 min. The developing solution was a PM thinner manufactured by Tokyo Ohka Kogyo Co., Ltd., Kawasaki, Japan. The developed SU-8 structure was used to mold the PDMS (Silpot 184; DuPont Toray Specialty Materials K.K.). The main liquid and curing agent of the PDMS were mixed in a ratio of 10:1 and degassed in a vacuum chamber until the air bubbles disappeared. The mixture was poured into the mold and again degassed in a vacuum chamber until there were no more bubbles. Afterwards, it was heat-cured at 90 °C for 30 min in an oven. After cooling to 25 °C or less, the PDMS was peeled off from the substrate. An inlet well and an outlet well were formed by punching holes having diameters of 1 mm. Finally, using PIB-20 (Vacuum Device Inc., Mito, Japan), the PDMS and glass were bonded in an air atmosphere at a pressure of 80 Pa, current of 30 mA, and plasma irradiation time of 30 s. Within 30 s after the hydrophilization, the PDMS and glass bonded together and were then heated on a hot plate at 95 °C for 5 min.

### 2.3. Microscopic Observation

Figure 4 shows an overview of the experimental setup, comprising an upright microscope (DM750, Leica Microsystems, Wetzlar, Germany), a microfluidic device, a syringe pump (MSPE-1, AS ONE Corporation, Osaka, Japan), and a high-speed CMOS camera (HAS-U2, DITECT Co., Ltd., Tokyo, Japan). The syringe pump was equipped with a stepping motor to control the pushing amount. In this experiment, different flow conditions of 1.5, 2.5, 8.0, 20 and 100 μL/min were used. A suspension of monodispersed polystyrene latex particles (Polysciences Inc., Warrington, PA, USA) with diameters of 3.0, 10.0 and 45.0 µm was used as the microscopic object. Because the size of the cells varied, we used a spherical object with a constant diameter. Although the size varied according to the cell type, an equivalent diameter of 10.0 µm was used considering the future applications using the cells. Then, 3.0 and 45.0 μm were added to confirm the difference in size. In addition, the number of particles per ml was adjusted to 4.6 × 10^6^ for diameters of 3.0 μm, 10.0 μm, and 4.6 × 10^5^ for diameters of 45.0 μm. Concentrations were adjusted using pure water. Using the above concentration as a standard, we confirmed in advance that the concentrations ranged from a 2-fold concentration to 100-fold, and confirmed from preliminary experiments that the concentration did not significantly affect the experimental results. However, in a previous study, the flow changed depending on the density in the case of red blood cells. The reason for this is that the higher the density of red blood cells, the more vigorous the agitation. Therefore, the results may vary if the concentration is extremely high [30].

Before starting the experiment, the inside of the microfluidic device was filled with pure water. Thereafter, the suspension was injected from the inlet well side of the microfluidic device using the syringe pump, and the behavior of the particles was observed. In this experiment, the time of passing through the chamber area of 800 μm × 800 μm was measured from the image information of the high-speed camera. As shown in the right side of Figure 4, the two boundary lines between the channels connecting the chambers were set as the start and goal lines.

## 3. Results

The experiments were performed at five different flow rates, and the chamber-passing time of 10 random samples was measured for each flow rate. Figure 5 shows the transit time for each flow rate. This graph is shown as a box-and-whisker plot to grasp the scattered data and to consider its application in component separation. In the case of the box-and-whisker plot, 50% of the total of the measured samples is in the area shown in the colored box. Moreover, the error bars indicate the minimum and maximum values; the x mark indicates the average value. However, for the particles with diameters of 45.0 μm, when the flow rate was less than 8.0 µL/min, the particles were too slow to reach the finish line and were thus trapped at the bottom of the chamber. Therefore, only the flow rates of 20 and 100 µL/min are shown in Figure 5C. Figure 6 displays a logarithmic graph with the horizontal axis representing the flow rate. The plot in Figure 5 shows the average value.

Figure 7 shows a captured image of the particles with diameters of 45.0 μm adhering to the chamber, gradually accumulating, and finally acquiring the state shown. Similar results were obtained for the polystyrene particles with diameters of 10.0 μm, and adhesion was confirmed when the flow rate was lowered. In the case of the polystyrene particles with diameters of 10.0 μm, no adhesion was observed at 1.5 µL/min; however, adhesion was confirmed at 1.0 µL/min. Owing to the deposition, the objects around it had difficulty flowing. This phenomenon is more likely to occur when the target concentration is high.

## 4. Discussion

This research aimed to contribute to component separation and cell trapping applications. Therefore, the considerations to obtain the necessary knowledge for these two applications are presented in this section. First, the results presented in Figure 5 and Figure 6 show that the transit time through the chamber decreased as the flow rate increased. However, there was no linear relationship between the velocity and transit time. For example, considering the results of 20 and 100 µL/min in Figure 5C, the passage time did not increase five times even though the flow rate was reduced to 1/5. Since the particles suffered resistance while flowing, it could be predicted that the flow velocity would not increase even if the flow rate was increased. Therefore, a sudden change in velocity occurred, as shown in the semi-logarithmic graph in Figure 7. In other words, when extracting only the liquid component from the suspension of particles or when recovering only the plasma component from the blood, the recovery efficiency was improved by increasing the flow rate. If the required liquid volume is large, the distance from the inlet well to the outlet well should be increased. However, it is unfeasible to continuously extract liquid. When a particle suspension was used, after a certain period of time, the particles were always extracted at a concentration higher than that before injection. Therefore, the microfluidic device could be used for concentration adjustment. Separation by increasing the flow rate was based on the resistance suffered by the particles, and the liquid, suffering no resistance, flowed quickly. Moreover, it was found that when the flow rate fell below the threshold, the particles completely stopped moving in the chamber. For example, in the case of a particle suspension with particle diameters of 45.0 µm or more, only the liquid component came out of the outlet at a flow rate of 8.0 µL/min or less. Although the amount of liquid obtained per unit of time was small, only the liquid component could be obtained continuously. Therefore, it was confirmed that the effect was obtained whether the flow rate increased or decreased.

Next, we confirmed the type of phenomenon occurring in the chamber. Since it was very difficult to numerically calculate the velocity vector in the chamber according to the particle diameter and flow rate using the finite element method, the experimental results were considered. Figure 8 shows the particle trajectories. The positions of the particles with diameters of 10.0 and 45.0 μm were plotted every 0.0333 s (equivalent to 30 fps) when the particles flowed at 20 μL/min. The particle position and the moving speed could be derived considering the plot interval. Four particles were randomly selected and color-coded, flowing from the upper left where the inlet well was located to the lower right where the outlet well was located. Particularly for 45.0 μm, the plot interval was short in the chamber area. Additionally, it was confirmed that velocity reduction in the chamber caused a separation phenomenon. Furthermore, when the flow rate decreased, the velocity of the particles also decreased; therefore, it was considered that the particles were deposited near the center of the chamber, as shown in Figure 7, owing to gravity and adhesion to the glass substrate. In addition, as can be seen from the particle behavior for each color in Figure 8, the velocity vector changed depending on the chamber and the trajectory through the channels before and after the chamber. As a result, the variation in transit time was affected.

Next, we considered whether it was possible to separate the particles by size and extract only the liquid component. Figure 9 presents a comparison of the transit times for each particle diameter. Figure 9A shows the flow rate of 8.0 µm/min, and Figure 9B shows the flow rate of 100 m/min. When the flow rate was the same, the larger the particle diameter, the greater the variation. Similar results were obtained under the other flow conditions. Theoretically, at the flow rate of 2.5 μm/min, it should have taken 1.92 s to fill the entire chamber with liquid. Afterward, for 0.39 s, only the liquid component did not contain particles with the diameter of 3.0 μm. At the flow rate of 2.5 μm/min, a group of particles having 10.0 μm diameter flowed for 0.0629 s after at least half of the 3.0 μm diameter particles passed through. Therefore, if the starting line was started simultaneously, the liquid, 3.0 μm in diameter and 10.0 μm in diameter, arrived in that order; thus, separation could be expected. On the other hand, at a flow rate of 100 μm/min, the diameters of 3.0 μm and 10.0 μm could not be separated because the transit time ranges overlapped. However, the 45.0 μm particles could be separated. It was confirmed that a favorable separation range existed depending on the flow rate conditions. An advantage of this method is that, when objects with small and large particle diameters are mixed, it is possible to separate objects with small particle diameters by optimizing the flow rate conditions. For example, in the case of blood, it is thought that the plasma can be separated first, then the liquid containing abundant red blood cells, and finally the liquid containing both red and white blood cells. Although it is not suitable for separating only white blood cells, it may be effective when collecting only the red blood cells.

Finally, a design was considered for experiments with the same channel size and half the chamber size of 400 μm × 400 μm. Figure 10 depicts the trajectory of the beads with diameters of 10.0 and 45.0 μm at 100 μL/min, plotted every 0.03333 s (30 fps). Even with a small chamber size, the particles slowed. In addition, when the chamber size was small, the plot interval was narrower for the particles with larger diameters. Comparing Figure 8 and Figure 10, there was no clear difference in the plot intervals. Even in applications where it was difficult to secure the number of chambers and areas, the effect could be realized by devising the design.

## 5. Conclusions

This study reported on component separation using a microfluidic device characterized most distinctively by its simple design comprising of only chambers and channels. It was proposed to be applied as a method for recovering liquids and small-sized particles. It was experimentally confirmed that the passage time through the chamber differed for each particle size, and the velocity of the particles decreased in the region near the center of the chamber. Based on the above results, the effect of the flow rate of the syringe on the separation performance was assessed. By reducing the flow rate below the threshold, the particles could be trapped, exhibiting a filtering behavior. Despite the submillimeter-order channel design, we were able to trap the 10.0 μm particles by controlling the flow rate. This effect was confirmed even when the chamber was smaller than the channel. Compared to other separation methods, the manufacturing cost is low, and the separation can be performed by optimizing only the flow rate parameters. We believe that it can be adapted according to the purpose of component separation, such as the size of the target object and the required amount of liquid. As it is possible to measure the pH of the liquid component and evaluate the contaminants, we hope that it will contribute to the early identification of the cause. We hope that this research will be useful, particularly for examinations in the medical and biotechnology fields in the future.

## Figures and Tables

**Figure 1 micromachines-14-00919-f001:**
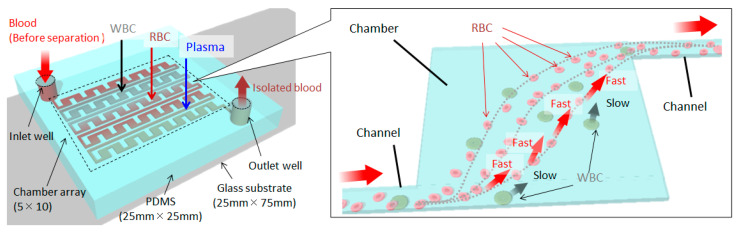
Design concept diagram. The left side represents the entire 25 mm × 25 mm PDMS fluidic device, integrated on a glass slide. Fifty chambers were connected in series, and the suspension flowed from the inlet well to the outlet well. The right side shows an enlarged view of a chamber. The larger the particle size, the slower it flowed through the chamber. Therefore, in the case of blood, plasma, red blood cells (RBCs), and white blood cells (WBCs) flowed in this order of decreasing speed.

**Figure 2 micromachines-14-00919-f002:**
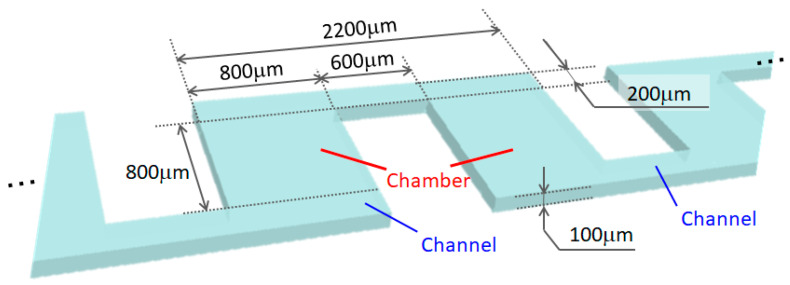
Design of the chamber and flow channel. The incoming suspension passed through 50 chambers from the inlet well to the outlet well. The chambers were 800 μm × 800 μm, and they were connected by 200 μm × 2200 μm channels. However, because the chambers were arranged in a grid pattern, the length of the flow path was more than 200 μm at some folded points. The overall height was 100 μm.

**Figure 3 micromachines-14-00919-f003:**
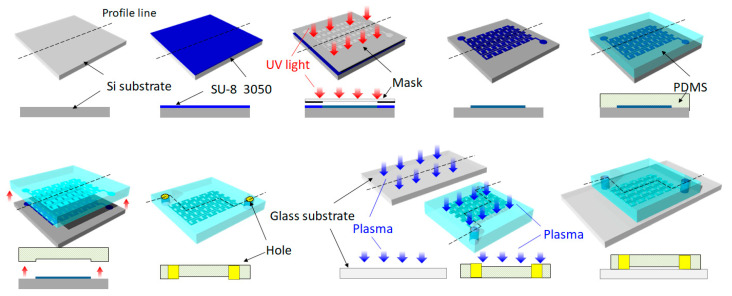
Fabrication was done by a general soft lithography process. Bird’s-eye and cross-sectional views are shown, and the profile line is shown in the bird’s-eye view. A mask with the channel design, shown in Figure 1 and Figure 2, was prepared and photolithography was performed. SU-8 3050 was used as a photoresist. Afterwards, PDMS was molded and holes for inlet and outlet wells were made. After plasma irradiation, glass and PDMS were bonded.

**Figure 4 micromachines-14-00919-f004:**
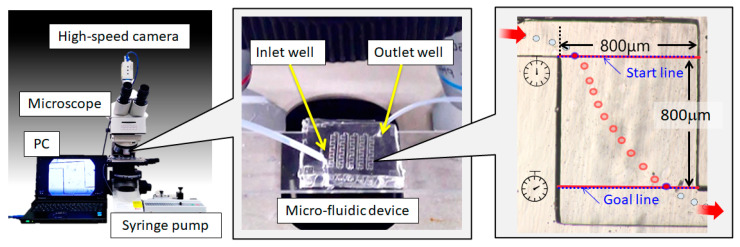
Experimental equipment. The left side shows an external view. It comprised a microscope, camera, PC for recording, and syringe pump. A microfluidic device was installed near the lens, and an enlarged photograph was shown at the center. The syringe was connected to the inlet well with a tube. The image on the right was obtained from the microscope. The time required to pass through an 800 μm × 800 μm chamber was measured. The start and goal lines shown in the figure were used as measurement standards. Red arrows indicate input and output directions.

**Figure 5 micromachines-14-00919-f005:**
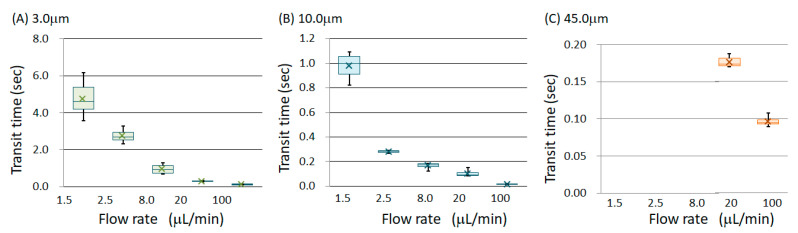
Transit time of polystyrene particles through the chamber for diameters of (**A**) 3.0 μm, (**B**) 10.0 μm, (**C**) 45.0 μm. The distribution at each flow rate is shown in a boxplot. When the diameter was 45.0 μm and the flow rate was less than 8.0 μL/min, the particles were too slow to reach the goal line.

**Figure 6 micromachines-14-00919-f006:**
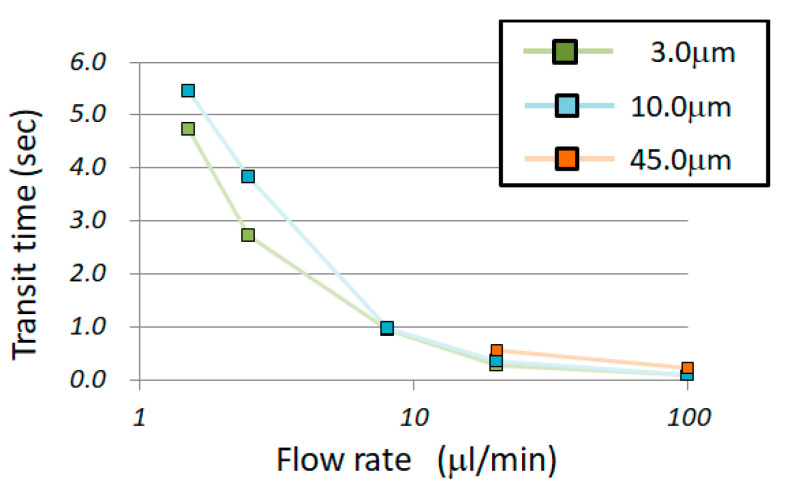
Correlation between flow rate and chamber passage time of polystyrene particles. Each plot is the average transit time.

**Figure 7 micromachines-14-00919-f007:**
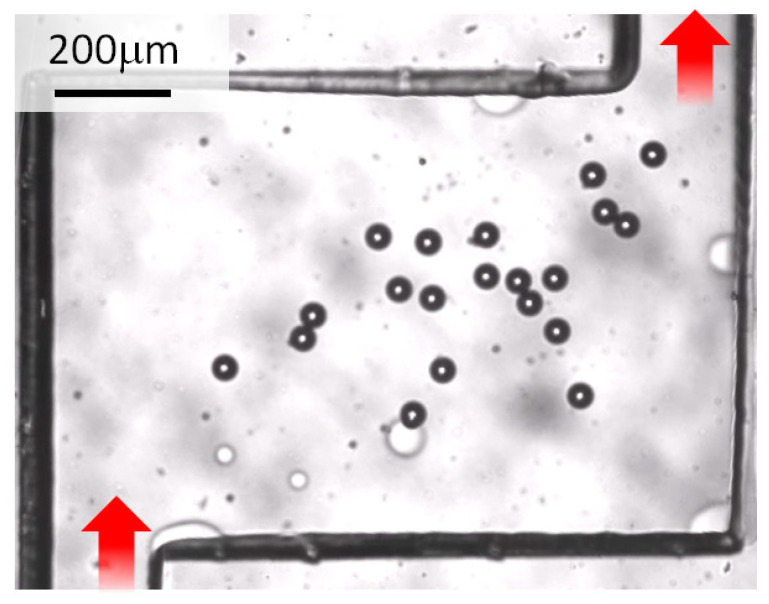
Results of particle trapping inside the chamber. Particles with diameters of 45.0 μm failed to reach the goal line and deposited when the flow rate was less than 8.0 μL/min. Red arrows indicate input and output directions.

**Figure 8 micromachines-14-00919-f008:**
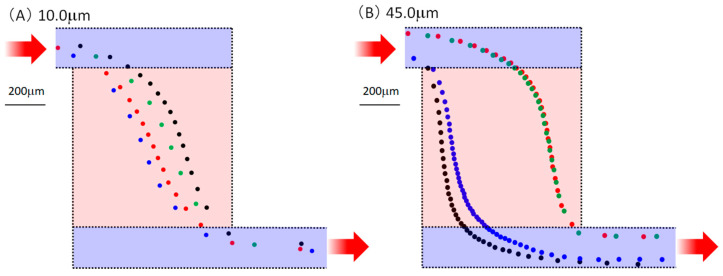
Behavior of particles inside the chamber. Red arrows indicate input and output directions. Positions of particles with diameters of (**A**) 10.0 and (**B**) 45.0 μm plotted every 0.0333 s (equivalent to 30 fps) when the particles flowed at 20 μL/min. And the plotted point is shown in the figure by a circle centered on it. Four random samples were selected and color-coded. In other words, the trajectory is a line connecting circles of the same color from upper left to lower right.

**Figure 9 micromachines-14-00919-f009:**
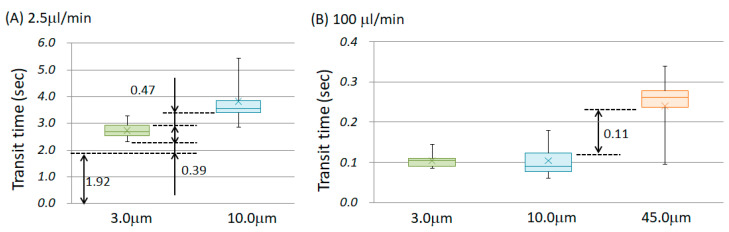
Comparison results of particle sizes and passage times at the same flow rate of (**A**) 8.0 μm/min and (**B**) 100 μm/min. 50% of the total of the measured samples is in the area shown in the colored box. The error bars indicate the minimum and maximum values; the x mark indicates the average value. The time interval between boxes is shown in the graph. For example, at the flow rate of 100 μm/min, particles with the diameter of 45.0 μm flowed after more than half of the particles with the diameter of 10.0 μm flowed.

**Figure 10 micromachines-14-00919-f010:**
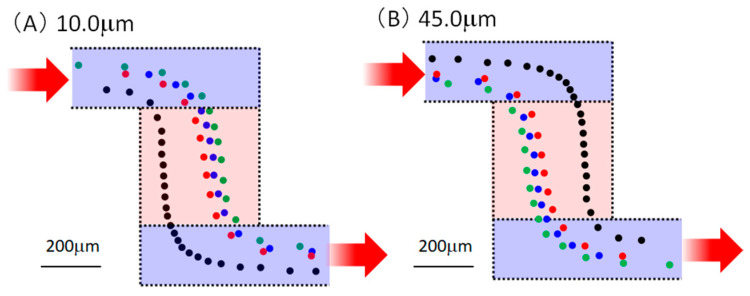
Confirmation of behavior in the chamber using a device w ith different chamber size. Red arrows indicate input and output directions. The position of particles with diameters of (**A**) 10.0 and (**B**) 45.0 μm was plotted every 0.0333 s (equivalent to 30 fps) when the particles flowed at 20 μL/min. And the plotted point is shown in the figure by a circle centered on it. Four random samples were selected and color-coded. In other words, the trajectory is a line connecting circles of the same color from upper left to lower right.

## Data Availability

The datasets used and/or analyzed during the current study are available from the corresponding author upon reasonable request.

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
