# Peer review of "Particle Size-Dependent Component Separation Using Serially Arrayed Micro-Chambers"

_micromachines, 2023, doi:10.3390/mi14050919_

Round 1
Reviewer 1 Report
The manuscript describes an interesting technique for cell separation using a novel microfluidic device. Here are some things which should be considered:
Broad comments:
1. The approach here uses particles with sizes 45 and 10 um. This size difference is considerable. How sensitive is this method for particles where sizes are not that high?
2. The paper discusses several issues like effect of concentration, extraction of only the liquid components without much background or results. More evidence should be provided for discussing these factors.
3. You have not discussed the phenomena which is causing this separation of particles. More insight into the science behind this technique will help the reader understand what is happening here. Showing differences in the trajectories of the 10 and 45 um particles will be beneficial.
4. The manuscript needs to be organized better. The pieces of discussions and potential applications of this technique are spread in different sections and are difficult to follow. for example lines 56-58 and 77-87 discuss manual pushing which is not used in the manuscript.
5. Proof read the manuscript for grammar and punctuation.
Specific comments:
1. Figures 5 and 6 can be combined. Using curve fitting to obtain an equation for data in Figure 5 can be beneficial.
2. It will be useful if you can draw a relation between transit time and particle size.
3. Figure 9 is missing.
4. Figure 8 does not add much value unless you use it to compare the trajectories of 10 and 45 um particles.
Author Response
Thank you very much for providing important comments. We are thankful for the time and energy you expended. We have significantly revised the manuscript according to the reviewers’ comments.
We hope that the reorganized content is in accordance with the comments.
Please refer to the PDF file for the answers to the questions.
I would like to thank you in advance for reviewing this submission and look forward to hearing from you.
Sincerely,
Mitsuhiro Horade

Reviewer 2 Report
In this manuscript, the authors investigated a method for particle separation using a simple series connection of chambers of the same shape via interconnecting channels. This design eliminates the need for a centrifuge and enables easy component separation on the spot without using a battery. Particles with a smaller size can be extracted more rapidly from the outlet. The separation tests can be conducted by manual pushing based on the flow rate and size of the object. However, i have several concerns before this manuscript can be accepted. Therefore, in its current form, revisions are needed.
1. How will the geometric size of the chamber and channel influence the sorting performance?
2. Are there clogging issues in the experiments? How do the authors avoid this problem?
3.How will the particle or cell’s densities affect the separation performance?
4.There is no Fig 9, but the authors discussed a lot about this Fig
5. Can this chip be reused again?
6. Comments on language:
- For example,
Lack punctuation marks (Line 176);
Author Response

(The authors gave the same response as above.)

Round 2
Reviewer 2 Report
The authors have addressed all my concerns, so I would like to recommend the paper for publication.